# The Impact of Next-Generation Sequencing Added to Multiplex PCR on Antibiotic Stewardship in Critically Ill Patients with Suspected Pneumonia

**DOI:** 10.3390/diagnostics14131388

**Published:** 2024-06-29

**Authors:** Kendall Kling, Chao Qi, Richard G. Wunderink, Chiagozie Pickens

**Affiliations:** 1Division of Infectious Diseases, Department of Medicine, Northwestern University Feinberg School of Medicine, Chicago, IL 60611, USA; 2Department of Pathology, Northwestern Memorial Hospital, Chicago, IL 60611, USA; 3Division of Pulmonary and Critical Care Medicine, Department of Medicine, Northwestern University Feinberg School of Medicine, Chicago, IL 60611, USA

**Keywords:** diagnostics, antibiotics, pneumonia, critical care, respiratory, molecular

## Abstract

Introduction: In patients with suspected pneumonia who are tested with respiratory culture and multiplex PCR, the potential added benefit of next-generation sequencing technologies is unknown. Methods: This was a single-center, retrospective study in which residual bronchoalveolar lavage (BAL) specimens were retrieved from hospitalized patients. We compared its research-use-only Respiratory Pathogen Illumina Panel (RPIP) results to culture and BioFire^®^ FilmArray Pneumonia Panel (BioFire^®^ PN) results from critically ill patients. Results: In total, 47 BAL specimens from 47 unique patients were included. All BAL samples were tested with culture and multiplex PCR. In total, 38 of the 47 BALs were consistent with a clinical picture of pneumonia per chart review. Additional testing of the 38 samples with the RPIP identified a new bacterium in 20 patients, a new virus in 4 patients, a new bacterium plus virus in 4 patients, and no additional organisms in 10 patients. In 17 (44.5%) of these patients, the RPIP results could have indicated an antibiotic addition. Compared with cultures, the RPIP had an overall sensitivity of 64% and specificity of 98%, with a 0% sensitivity for fungus and 14% sensitivity for mycobacteria. Compared with BioFire^®^ PN, the RPIP was 70% sensitive and 99% specific, with a 74% sensitivity for bacteria and 33% sensitivity for viruses. The RPIP was 29% more sensitive for HAP/VAP bacterial targets compared with CAP. Conclusions: Emerging NGS technologies such as the RPIP may have a role in identifying the etiology of pneumonia, even when patients have BAL culture and multiplex PCR results available. Similar to prior studies evaluating RPIP, our study showed this platform lacked sensitivity when compared with cultures, particularly for fungi and mycobacteria. However, the high specificity of the test can be leveraged when clinicians are seeking to rule out certain infections.

## 1. Introduction

Pneumonia is the most common infectious cause of death in the United States [1], and timely administration of appropriate antibiotics is associated with improved mortality [2,3]. Clinicians are encouraged to tailor antibiotics to the pathogens detected in diagnostic studies to optimize antibiotic stewardship and minimize the risk of adverse effects associated with antibiotics [2,4]. Respiratory culture, the gold-standard diagnostic test for pneumonia, is limited by prolonged turnaround time, reduced sensitivity in patients receiving antimicrobial therapy, and an inability to detect fastidious or non-cultivatable organisms. The causative organism in pneumonia is only detected in 30% of hospitalized patients [5], and bronchoalveolar lavage (BAL) culture positivity can be as low as 5% in patients receiving antibiotics [6]. Culture-independent rapid diagnostic tests (RDTs) have been developed to address the limitations of cultures and improve antibiotic stewardship. The FDA-approved commercially available molecular RDTs for lower respiratory tract infection detect organisms commonly implicated in community-acquired and nosocomial pneumonia but may miss opportunistic infections. One such panel is the BioFire^®^ FilmArray^®^ Pneumonia Panel (BioFire^®^ PN, BioFire Diagnostics, Salt Lake City, UT, USA), a syndromic multiplex nested PCR that can detect 26 pathogens (18 bacterial and 8 viral) and seven genetic resistance markers from sputum, BAL, or endotracheal aspirates with a roughly 60 min turnaround time [7]. A multi-center study of this panel suggested the panel had higher positivity rates when compared with cultures [8].

Given the high rates of culture-negative pneumonia [9] and the increasing number of hospitalized patients at risk for lower respiratory tract infections from uncommon organisms [10], interest in the application of next-generation sequencing (NGS) to clinical practice is growing. Recently, a prospective, randomized controlled trial of antibiotic prescribing guided by NGS compared with cultures was conducted on ICU patients. The trial reported that 48% of the results informed antimicrobial prescribing changes (22% escalation; 26% de-escalation) [11]. The value of NGS in identifying the etiology of pneumonia when added to standard-of-care testing plus multiplex PCR is unclear. This creates a knowledge gap that needs to be addressed to optimize resource utilization and patient care.

NGS, though still in its infancy in clinical microbiology laboratories, can potentially improve BAL diagnostics by detecting fastidious or difficult-to-grow organisms, detecting organisms in the setting of prior antimicrobial treatment, and assisting in the diagnosis of polymicrobial infections [12]. Currently, few commercially available NGS platforms for clinical microbiology laboratories are available [13]. The Respiratory Pathogen Illumina Panel (RPIP, Illumina, San Diego, CA) is a targeted NGS panel designed for BAL specimens that can detect 282 pathogens (187 bacteria, 42 viruses, and 53 fungi). To understand the potential added benefit of a comprehensive, culture-independent diagnostic test in hospitalized patients undergoing BAL, we performed a retrospective study of the RPIP panel. We hypothesized that potential pathogens that would impact antibiotic therapy could be detected when results from the RPIP are added to the results from BAL cultures and the BioFire^®^ PN.

## 2. Materials and Methods

This was a single-center, retrospective study conducted at Northwestern Memorial Hospital from August to September 2021. This study received IRB approval from Northwestern University (STU00218511). Residual BAL specimens from hospitalized patients were retrieved from the microbiology lab once all clinical testing was completed. All specimens were de-identified and assigned a unique code, archived, and frozen. Clinical data were obtained from the electronic medical record and Enterprise Data Warehouse. Hospital-acquired pneumonia was defined as pneumonia onset 48 h or more after hospital admission, and ventilator-associated pneumonia (VAP) was defined as pneumonia onset after 48 h of intubation [14]. Clinical adjudication by chart review was performed by an infectious disease physician to confirm the diagnosis of pneumonia. Microbiology results were obtained from our laboratory information system (LIS).

### 2.1. Extraction and Library Preparation

Total nucleic extraction was performed with the ZymoBINOMICS DNA/RNA MiniPrep Kit DNA (Zymo Research, Irvine, CA, USA). Nucleic acid was extracted with bead beading and lysis buffer and eluted with ultra-pure nuclease-free water. DNA was quantified via a Qubit fluorometer (Thermo Fisher Scientific, Waltham, MA, USA). RNA was not quantified per manufacturer protocol. Library preparation was carried out via the manufacturer’s protocol via the Respiratory Pathogen ID/AMR Enrichment (RPIP) kit via the Illumina RNA prep with enrichment tagmentation protocol. Positive controls included a Microbiologics respiratory control panel (catalog number 8247) with 22 targets (Microbiologics, Saint Cloud, MN, USA) and a viral control via the T7 Phage Stock (Microbiologics, Saint Cloud, MN, USA) in Universal Transport Medium (Quidel, San Diego, CA, USA). The negative control was ultra-pure nuclease-free water. The total extracted nucleic acid and controls underwent cDNA synthesis via a two-stage reaction. The cDNA then underwent tagmentation via enrichment-bead-linked transposomes. The fragments were purified and amplified, and index adapter sequences were added for indexing and clustering. Libraries were then normalized and combined in three-plex enrichment to standard concentrations. Pooled libraries were then hybridized with the enrichment oligonucleotides; captured with magnetic beads; and then washed, eluted, and amplified. The enriched library was then analyzed with a fragment analyzer (5200 Fragment Analyzer System, Agilent, Santa Clara, CA, USA) with a goal average peak length of 400–500 base pairs. Libraries were diluted to a final loading concentration of 10 picomolar, loaded to and sequenced by the MiSeq System (Illumina), and then analyzed via the Explify RPIP Data Analysis pipeline via Illumina BaseSpace. Sequencing was performed with 150 bp paired-end read sequencing.

### 2.2. Determination of Agreement

Complete agreement was defined as positive or negative results that were completely concordant between testing modalities. Organisms on BAL that are considered normal respiratory tract microbiota are inconsistently reported by our institution’s microbiology lab and, therefore, not included in the analysis of operating characteristics (note, these organisms *were* included when available for evaluating the potential impact on antimicrobial therapy). These organisms include *Bacillus* species, coagulase-negative *Staphylococcus*, *Corynebacterium* species, *Haemophilus* species (not *H. influenzae*), *Micrococcus* species, *Neisseria* species, *Rothia* species, and viridans group *Streptococci*.

The results of the RPIP were also directly compared with the BioFire^®^ PN. Only targets on the BioFire^®^ PN were included when evaluating the test’s operating characteristics. Organisms reported in our BAL cultures and targets on the BioFire^®^ PN are included in Appendix A. The 282 organisms targeted by the RPIP are available online. Although the RPIP panel does report genotypic markers of antimicrobial resistance, our study did not evaluate this portion of the test given the uncertain interpretation of discordant results.

### 2.3. Statistical Analysis

Statistical analysis was performed using RStudio version 4.3.1 and Microsoft Office Excel (2021). Continuous variables that were not evenly distributed were reported as medians with the first and third interquartile ranges in brackets. Continuous variables were compared using the Kruskal–Wallis test with Bonferroni correction for multiple comparisons. Categorical variables were compared using Fisher’s exact test. The alluvial plot was generated using the *ggalluvial* package in RStudio.

## 3. Results

A total of 47 BALs from 47 unique patients were included in this study. The clinical characteristics of the cohort are reported in Table 1. A total of 38 (80%) members of the cohort had suspected pneumonia. In total, 11 (23%) patients had CAP, and 27 (57.4%) had HAP/VAP. Nine (19%) of the patients did not meet the clinical criteria for pneumonia based on chart review. Thirteen (27.7%) of the patients were immunocompromised. Of the nine cases where pneumonia was not suspected by the clinical team, eight had negative cultures, and all nine had negative BioFire^®^ PN results. The RPIP results were negative in three of the nine non-pneumonia cases; positive for *Neisseria flavescens*, *Veillonella*, *Capnocytophaga*, *Prevotella*, *Rothia*, *Campylobacter consicus* in four; and positive for HSV-1 and CMV in two cases.

### 3.1. General Overview of Organism Detection

Positive controls detected the anticipated targets, and negative controls were negative. The operating characteristics of the panel are demonstrated in Table 2. The organisms identified by all testing methods are listed in Appendix A. RPIP detected a total of 99 organisms, 34 of which may be considered pathogens, and 65 microorganisms of uncertain pathogenicity in the lung.

### 3.2. RPIP Compared with Culture and BioFire^®^ Pneumonia Panel

The overall sensitivity of RPIP compared with cultures was 63.9%, and specificity was 98.1%. Sensitivity was lower for mycobacteria at 14.3% (1/7) and was 0% (0/4) for fungus. Complete agreement between the cultures and RPIP was observed in 24 (51%) of the 47 BALs (Figure 1). Of the 23 BALs with incomplete agreement, 9 (39.1%) were culture-negative and RPIP-positive results; 7 (30.4%) were culture-positive and RPIP-negative; and 7 (30.4%) had mixed results. Notably, the cultures detected three Aspergillus species and one Rhizopus, all of which were missed by the RPIP. The cultures also detected seven mycobacteria that were missed by the RPIP; however, the RPIP detected one *Mycobacterium avium* complex in an acid-fast-smear-positive patient who was negative for AFB culture. The overall sensitivity of the RPIP compared with the BioFire^®^ Pneumonia Panel was 69.0%, and specificity was 98.6%. The sensitivity for viruses was 33.3% in this study. Complete agreement between BioFire^®^ and the RPIP was observed in 35 (74.5%) of the 47 BALs (Figure 2). There was no correlation between semi-quantitation on the BioFire^®^ PN and the determination of false-positivity and -negativity (Table 3).

### 3.3. RPIP for Resolving Discrepant Culture and BioFire^®^ PN

The ability of the RPIP to resolve discrepant culture and BioFire^®^ Pneumonia Panel results was assessed. A total of 13 (27.7%) of the 47 BALs had discrepant culture and BioFire^®^ results, of which 12 (92.3%) were Biofire^®^-positive and culture-negative. Of these 12 BioFire^®^-positive, culture-negative results, the RPIP agreed with the culture in nine (75%) cases and with BioFire^®^ in three (25%). In the single BioFire^®^-negative, culture-positive BAL, the RPIP was concordant with BioFire^®^. Of the nine cases in which BioFire^®^ was positive but the culture and RPIP were negative, further testing was performed to adjudicate the result. Four (44.4%) were determined to be true-positive BioFire^®^ results, three (33.3%) were determined to be false-positive BioFire^®^ results, and two (22.2%) were undetermined given a lack of other ancillary testing.

### 3.4. RPIP for CAP versus HAP/VAP

RPIP was 29% more sensitive for non-mycobacterial bacterial targets for HAP/VAP as compared with CAP (Table 2). For BioFire^®^ PN, RPIP performed similarly overall for CAP versus HAP/VAP for bacterial targets. Of the 11 CAP BALs, four (36.4%) had complete agreement with the RPIP, the culture, and BioFire^®^ PN. The RPIP missed a total of six pathogens in patients with CAP, including three *M. avium*, one SARS-CoV-2, one *K. pneumoniae*, and one Rhizopus. The RPIP detected a new pathogen in 2 (18%) of the 11 cases, including one *M. avium* and one Adenovirus. Of note, these organisms, particularly Mycobacterium and Rhizopus, are often not pathogens in immunocompetent individuals.

### 3.5. Therapeutic Impact of RPIP

We examined the potential impact of RPIP results on antibiotic therapy for 38 cases of suspected pneumonia. All participants in this study received a BAL culture and multiplex PCR; therefore, we investigated the potential benefit of the RPIP result added to these tests. All identified organisms, including organisms of unclear pathogenicity, were regarded as potential pathogens warranting treatment. This approach was used to capture the maximum potential impact that the RPIP could have on antibiotic escalation. We defined two groups of potential antibiotic change:No change: The antibiotic regimen for the identified organisms upon culture and/or mPCR would sufficiently cover the organisms identified by the RPIP.Additional therapy: The organism identified by the RPIP would not be covered by the antibiotics used for the culture- or mPCR-identified organisms. If no organisms were identified by cultures or mPCR, the patient was assumed to be on no antibiotics.

In 10 (26.3%) cases, no new organisms were identified by the RPIP and, therefore, no additional antibiotic would be indicated. Four culture- and BioFire^®^-negative cases had a result from the RPIP that could potentially lead to an antibiotic addition (Figure 3). The specific antibiotic changes for each case are listed in Appendix A.

## 4. Discussion

Commercial targeted NGS platforms are potentially attractive options for advanced diagnostics for clinical microbiology laboratories due to their ease of analysis compared with non-targeted NGS, but they are currently hampered by high costs, lengthy preparation times, and uncertain analytic performance [15]. We compared the RPIP to culture and BioFire^®^ Pneumonia Panel results to elucidate its potential benefit to routine diagnostics.

To the best of our knowledge, this study is the first to directly compare the operating characteristics of the RPIP to the BioFire^®^ PN. In our study, we found that the RPIP platform had a better yield for cases of HAP/VAP compared with CAP. One potential explanation for this is that HAP/VAP more commonly has a positive culture compared with CAP. The RPIP had reduced sensitivity when compared with both cultures and the BioFire^®^ PN while specificity was high. Compared with cultures, the sensitivity of the RPIP for the detection of fungal and mycobacterial organisms was particularly low.

Our study confirms the previously reported operating characteristics of the RPIP, including the limited ability of the assay to detect fungal and mycobacterial targets [15,16,17]. We noted potential cross-reactivity in the RPIP analysis possibly due to near-neighbor effects. For example, a BAL that grew *Mycobacterium avium* complex was identified as *M. kansasii* by the RPIP. Cross-reactivity has been reported previously, as Gaston et al. reported *M. fortuitum* and *M. tuberculosis* cross-reactivity, as well as *S. mitis* and *S. pneumoniae* [15]. A key feature of the RPIP is its ability to enrich pathogens on their panel; however, studies comparing the RPIP to shotgun metagenomics have found no benefit to this approach, which indicates that this technology could be further optimized [15].

When the RPIP was used to resolve the discrepant results between BAL cultures and BioFire, we found that the RPIP results were more consistently concordant with cultures. One possible explanation is that, as a real-time PCR-based assay (which is designed to detect targets of 75–150 base pairs) [18], BioFire^®^ PN detects shorter genomic DNA fragments than the RPIP. Therefore, in theory, BioFire^®^ PN could have a higher likelihood of detecting disintegrated microorganisms.

In hospitalized patients with suspected pneumonia who have a BAL culture and multiplex PCR test already performed, the RPIP will often detect additional organisms. In our cohort of patients who underwent BAL culture and multiplex PCR testing, up to 45% of patients would potentially require additional antibiotic therapy based on RPIP results. Because of the difficulty in differentiating a true pathogen from a commensal, we included all organisms as potential pathogens in this analysis. Many of these patients had oral microbiota and latent viruses detected; many times, these are not treated in clinical practice. Taken together, RPIP may have an added benefit for detecting the etiology of infections in patients tested with negative cultures and multiplex PCRs. While not useful as a screening test for pneumonia, the high specificity of the RPIP can be utilized if a false-positive is suspected from routine testing.

Our study has limitations. First, we had a relatively small sample size of 47. This precludes our findings from being generalizable, and larger prospective studies of the platform should be conducted. Second, our study had low numbers of viruses (*n* = 3), mycobacteria (*n* = 8), and fungi (*n* = 6), limiting our power to assess the RPIP’s ability to detect these organisms. In addition, most of our cohort (71%) were HAP/VAP, potentially skewing the pre-test probability to more bacterial targets, rather than viral, fungal, or mycobacterial.

In conclusion, the RPIP has the potential to expand the library of organisms detectable by clinical microbiology laboratories—including non-cultivable organisms such as *Coxiella burnetii* and *Chlamydia psittaci*. At present, these emerging technologies cannot replace cultures given their lack of sensitivity for fungal and mycobacterial pathogens. This may be less relevant to community-acquired infections, as fungal and mycobacterial organisms are uncommon causes of CAP. Rapid multiplex panels like the BioFire^®^ Pneumonia Panel also appear to offer higher sensitivity for viral targets and offer results much faster and with less hands-on time. We feel the RPIP panel could have potential use in cases of suspected pneumonia where cultures and RDTs are negative or in cases of positive results from either test but a lack of improvement in the patient’s clinical course. The RPIP cannot be reliably recommended to resolve discordant culture and BioFire^®^ PN results given the lower sensitivity of the RPIP compared with PN. In the future, NGS technologies could benefit clinical microbiology laboratories by obtaining FDA approval by streamlining protocols to shorten preparation times, reducing the cost of reagents, and improving sensitivity for fungal and mycobacterial targets by optimizing extraction methods. In addition, prospective studies are needed to address the clinical impact of microorganisms detected by these emerging technologies.

## Figures and Tables

**Figure 1 diagnostics-14-01388-f001:**
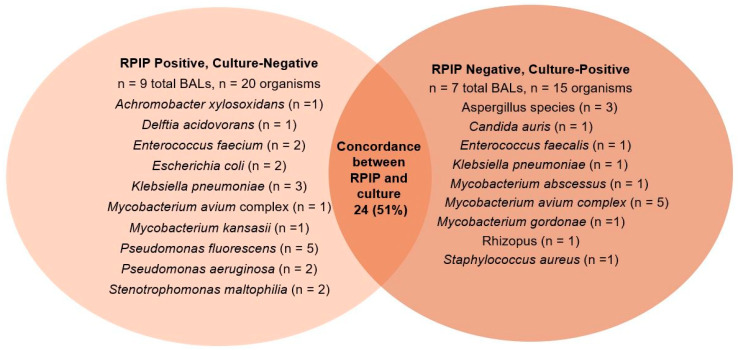
Concordance between RPIP and cultures, with discordant organisms listed in each circle.

**Figure 2 diagnostics-14-01388-f002:**
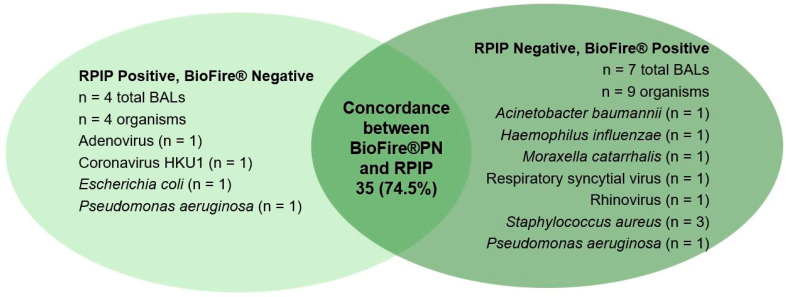
Concordance between RPIP and BioFire^®^PN, with discordant organisms listed in each circle.

**Figure 3 diagnostics-14-01388-f003:**
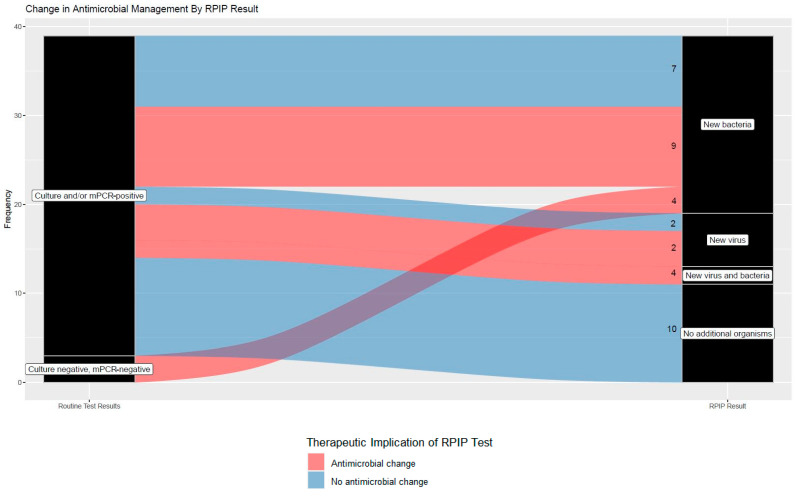
Alluvial plot of antibiotic changes that may be made based on RPIP results.

**Table 1 diagnostics-14-01388-t001:** Clinical characteristics and outcomes of the study cohort. Note that some datapoints were unavailable for participants, when this was the case the total *n* of the cohort is listed in parenthesis as (*n* = ).

	CAP (*n* = 11)	HAP/VAP (*n* = 27)	Non-Pneumonia (*n* = 9)	*p*
Age	57 [36.5, 72]	63 [42, 69.5]	67 [56, 82]	0.44
Gender,% female	63.6%	33.3%	33.3%	0.18
CCI	2.5 (*n* = 8)	4 (*n* = 26)	4 (*n* = 9)	0.24
Time from hospitalization to BAL	0 [0, 0.5]	10 [6.5, 15.5]	4 [1, 6]	0.00
% Culture-negative	36.4%	51.9%	88.9%	0.04
% BioFire-negative	90.0%	36.4%	100%	0.00
Peripheral leukocyte count (on day of BAL)	11.3 [6.4, 12.4] (*n* = 7)	12.2 [8.9, 17.9]	8.6 [8.2, 13.1]	0.44
Max daily temperature	98.4 [97.5, 99.5]	99.3 [98.5, 101.1]	99.6 [99.0, 100.0]	0.14
Hospital LOS	4 [0, 11.5]	36 [19, 50]	25 [11, 27]	0.00
BAL WBC	416 [258, 1065] (*n* = 10)	2006 [563, 4400] (*n* = 24)	792 [96, 4565] (*n* = 8)	0.67
BAL % neutrophils	85 [67, 95] (*n* = 9)	83 [45, 96] (*n* = 24)	71 [35, 91] (*n* = 8)	0.76
In-hospital mortality	27.3%	40.7%	22.2%	0.53

**Table 2 diagnostics-14-01388-t002:** Operating characteristics of RPIP compared with cultures and BioFire^®^ Panels.

Analytical Performance of RPIP Compared to Culture and BioFire^®^ Pneumonia Panel								
	TP	FP	TN	FN	Sensitivity	Specificity	PPV	NPV
**RPIP vs. Culture (All, *n* = 47, Total)**	23	2	104	13	63.9%	98.11%	92.00%	88.9%
RPIP vs. Culture (Bacterial)	22	1	23	2	91.7%	95.50%	95.65%	91.3%
RPIP vs. Culture (Mycobacterial)	1	1	39	6	14.3%	97.50%	50.00%	86.7%
RPIP vs. Culture (Fungal)	0	0	42	5	0.0%	100.0%	0.0%	89.4%
**RPIP vs. BioFire^®^ PN (All, *n* = 47, Total)**	21	1	70	9	70.0%	98.6%	95.5%	88.6%
RPIP vs. BioFire^®^ PN (Bacterial)	20	1	26	7	74.0%	96.3%	95.2%	78.8%
RPIP vs. BioFire^®^ Viral	1	0	44	2	33.3%	100.0%	100.0%	95.7%
	**TP**	**FP**	**TN**	**FN**	**Sensitivity**	**Specificity**	**PPV**	**NPV**
**RPIP vs. Culture (CAP, *n* = 11, Total)**	3	1	25	5	37.5%	96.2%	75.0%	83.3%
RPIP vs. Culture (Bacterial)	2	0	8	1	66.7%	100.0%	100.0%	88.9%
RPIP vs. Culture (Mycobacterial)	1	1	7	3	25.0%	87.5%	50.0%	70.0%
RPIP vs. Culture (Fungal)	0	0	10	1	0.0%	100.0%	0.0%	90.9%
**RPIP vs. BioFire^®^ (CAP, *n* = 11 Total)**	2	0	19	1	66.7%	100.0%	100.0%	95.0%
RPIP vs. BioFire^®^ PN (Bacterial)	1	0	9	1	50.0%	100.0%	100.0%	90.0%
RPIP vs. BioFire^®^ Viral	0	1	10	0	100.0%	100.0%	100.0%	100.0%
	**TP**	**FP**	**TN**	**FN**	**Sensitivity**	**Specificity**	**PPV**	**NPV**
**RPIP vs. Culture (HAP/VAP, *n* = 27)**	20	1	52	8	71.4%	98.1%	95.2%	86.7%
RPIP vs. Culture (Bacterial)	20	1	6	1	95.2%	85.7%	95.2%	85.7%
RPIP vs. Culture (Mycobacterial)	0	0	23	3	0.0%	100.0%	0.0%	88.5%
RPIP vs. Culture (Fungal)	0	0	23	4	0.0%	100.0%	0.0%	85.2%
**RPIP vs. BioFire^®^ (HAP/VAP, *n* = 27)**	19	1	33	8	69.2%	97.1%	94.7%	80.5%
RPIP vs. BioFire^®^ (Bacterial)	19	1	8	6	76.0%	88.9%	95.0%	57.1%
RPIP vs. BioFire^®^ Viral	0	0	25	2	0.0%	100.0%	0.0%	95.3%

**Table 3 diagnostics-14-01388-t003:** Correlation between semi-quantitation on the BioFire^®^ PN and determination of false-positivity and -negativity.

Case No.	Positive BioFire Result	Quantity (DNA Copies/mL)	Additional Testing	Final Determination
1	*Acinetobacter baumannii*	10^5^	Repeated negative BAL testing.	False-positive BioFire^®^.
32	*Acinetobacter baumannii*	10^4^	Positive *A. baumannii BAL* culture 2 months later.	True-positive BioFire^®^.
39	*Haemophilus influenzae*	10^4^	*H. parainfluenzae* was detected with RPIP.	Undetermined.
46	*Moraxella catarrhalis*	10^5^	No other data to support.	Undetermined.
11	*Pseudomonas aeruginosa*	10^4^	*P. aeruginosa* grew in BAL culture 5 days later.	True-positive BioFire^®^.
7	*Staphylococcus aureus*	10^5^	*S. aureus* grew in BAL culture 3 days prior.	True-positive BioFire^®^.
15	*Staphylococcus aureus*	10^4^	*S. aureus* grew in BAL culture 20 days prior.	True-positive BioFire^®^.
45	*Staphylococcus aureus*, mecA/MREJ	10^6^	Cepheid MRSA PCR BAL was negative same day.	False-positive BioFire^®^.
18	*Staphylococcus aureus*, mecA/MREJ	10^4^	Cepheid MRSA PCR from nares was negative the day prior.	False-positive BioFire^®^.

## Data Availability

Data and materials will be made available upon reasonable request.

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
