# Peer review of "The Impact of Next-Generation Sequencing Added to Multiplex PCR on Antibiotic Stewardship in Critically Ill Patients with Suspected Pneumonia"

_diagnostics, 2024, doi:10.3390/diagnostics14131388_

Round 1
Reviewer 1 Report
Comments and Suggestions for Authors
The study evaluates next-generation sequencing (NGS) for diagnosing pneumonia in critically ill patients by comparing the Respiratory Pathogen Illumina Panel (RPIP) to traditional culture and BioFire® FilmArray Pneumonia Panel (BioFire® PN). Using bronchoalveolar lavage (BAL) specimens from 47 patients, RPIP identified new pathogens in some cases and could influence antibiotic treatment in 44.5% of pneumonia cases. RPIP showed 64% sensitivity and 98% specificity compared to culture, and 70% sensitivity and 99% specificity compared to BioFire® PN, with better sensitivity for hospital-acquired (HAP/VAP) than community-acquired pneumonia (CAP). However, RPIP had limited sensitivity for fungi and mycobacteria. The study suggests RPIP is useful for ruling out infections due to its high specificity but is less effective as a standalone diagnostic tool due to lower sensitivity. The manuscript effectively communicates the main findings of the study. However, there are some recommendations that increase the quality of the manuscript.
1. The introduction section needs improvement as the research gaps are not clearly identified in the manuscript.
2. The study's relatively small sample size limits the generalizability and statistical power of the findings.
3. In Table 3, the value attributes for quantity are not understandable.
4. A more thorough discussion is required for the study.
Author Response
Reviewer 1 Comment
The study evaluates next-generation sequencing (NGS) for diagnosing pneumonia in critically ill patients by comparing the Respiratory Pathogen Illumina Panel (RPIP) to traditional culture and BioFire® FilmArray Pneumonia Panel (BioFire® PN). Using bronchoalveolar lavage (BAL) specimens from 47 patients, RPIP identified new pathogens in some cases and could influence antibiotic treatment in 44.5% of pneumonia cases. RPIP showed 64% sensitivity and 98% specificity compared to culture, and 70% sensitivity and 99% specificity compared to BioFire® PN, with better sensitivity for hospital-acquired (HAP/VAP) than community-acquired pneumonia (CAP). However, RPIP had limited sensitivity for fungi and mycobacteria. The study suggests RPIP is useful for ruling out infections due to its high specificity but is less effective as a standalone diagnostic tool due to lower sensitivity. The manuscript effectively communicates the main findings of the study. However, there are some recommendations that increase the quality of the manuscript.
The introduction section needs improvement as the research gaps are not clearly identified in the manuscript.
Response: Thank you for this comment, we agree with your assessment and have expanded the introduction. See lines 57-62.
Reviewer 1 Comment
The study's relatively small sample size limits the generalizability and statistical power of the findings.
Response: Thank you for this feedback. We agree that the sample size is a limitation of this study. We have added this to the discussion. See lines 251-253.
Reviewer 1 Comment
In Table 3, the value attributes for quantity are not understandable.
Response: We thank the reviewer for this observation. We agree and have added units to the Table. See Table 3 on page 7.
Reviewer 1 Comment
A more thorough discussion is required for the study.
Response: We agree, there were several discussion points that could be elaborated upon. We have added to the discussion while trying to remain within the limits of the word count. See lines 234-239, and 219-221.
Reviewer 2 Report
Comments and Suggestions for Authors
The research idea and its implementation are excellent. Objective - comparison of methods for identification of pneumonia infectious agents is of great importance for science and clinical practice.
The study rationale, research methods, results and discussion are presented very well. The conclusions are adequate.
Very good research was done.
I have no critical comments or concerns.
Author Response
Reviewer 2 Comments
The research idea and its implementation are excellent. Objective - comparison of methods for identification of pneumonia infectious agents is of great importance for science and clinical practice.
The study rationale, research methods, results and discussion are presented very well. The conclusions are adequate.
Very good research was done.
I have no critical comments or concerns.
Response: Thank you for taking time to review our submission.